# The relationship between physical fitness and competitive performance of Taekwondo athletes

Rui Liu[1], Lumin He[2]*

1 Department of Physical Education, Ocean University of China, Qingdao, China, 2 Affiliated Athletics School of Beijing Sport University, Beijing, China

* helumin@bsu.edu.cn

## Abstract

The competition and physical fitness test results of the 2020 National Taekwondo Championship Series were analyzed using curve fitting, linear regression, and other statistical methods. As far as we know, it is the first Taekwondo competition that uses physical fitness test (PFT) scores as the 8-in-4 selection criteria. The results show that the probability of the final total score of the series of championships entering the top 8 or top 3 is exponentially related to PFT results. It finds that athletes with better PFT scores are more likely to enter the quarterfinals. Among athletes entering the semifinals, the athlete with the best physical fitness has the greatest probability of winning the championship. The difference in physical fitness between athletes is mainly reflected in the 30-meter sprint. Overall, the competitive performance of professional Taekwondo athletes is significantly positively correlated with their physical fitness, especially for female Taekwondo athletes. Through the results obtained, it is concluded that Taekwondo athletes need to strengthen physical training, specifically enhancing the explosive power.

## 1. Introduction

Taekwondo originated in the Korean peninsula, which is famous for its flexible kicking techniques, and has been an Olympic combat discipline since 2000. In order to guide the training and competition more scientifically, reasons affecting the achievements of Taekwondo athletes have been studied extensively during the past few decades [1–5]. In general, performance in taekwondo is considered to be determined by a competitor's technical, tactical, psychological, physical and physiological characteristics [1, 2].

Since the establishment of the World Taekwondo Federation (WTF) in 1973, the competition rules of Taekwondo have been revised more than 40 times [6]. In competitions conducted according to the pre-2015 rules, Taekwondo athletes did not generally apply difficult technical approaches, but preferred to use secure frontal skills to win the game [7], which is why the roundhouse kick was the most popular means of scoring [8]. However, following the changes in competition rules, Taekwondo competition tactics now have offensive tactics as the main approach and counter-attack tactics as the secondary approach [9]. As an important

**Data Availability Statement:** All the data used in our paper can be found on the website of Chinese Taekwondo Association (http://www.taekwondo.org.cn/), however, the data in pdf files are write in

Chinese, we have translated it and included the English version as a Supporting Information file.

**Funding:** This study was supported by grant 201915002 from the Ocean University of China (Rui Liu received), and the grant "Elite coaches double hundred training plan implementation measures" from the General Administration of Sport of China. (Lumin He reviewed). The funders had no role in study design, data collection and analysis, decision to publish, or preparation of the manuscript.

**Competing interests:** The authors have declared that no competing interests exist.

foundation of athletes' competitive ability, physical fitness has also become increasingly prominent with the rule changes [10, 11].

In terms of physiological characteristics, it has been found that elite taekwondo athletes tend to possess low levels of body fat [12, 13], moderate to high levels of cardio-respiratory fitness [14] and high levels of both aerobic and anaerobic physical fitness [15], while muscle strength is often not a key role [16]. In terms of specific physical fitness characteristics, successful taekwondo athletes were found to possess significantly higher maximum running speed [12], better performance on 30m run, counter movement jump, moving sideways and walking backwards [17]. Additionally, research on the physical fitness test (PFT) of junior taekwondo athletes revealed that the power of lower extremities, strength and endurance are of great importance to the sports result [4, 18]. However, the aforementioned studies are limited by the fact that the PFTs performed were all conducted during training, because there is a big difference between training and competition modes for athletes' psychological and physical conditions [15, 19, 20]. Therefore, further study on the relationship between PFT and athletes' competition performance is still required.

Being aware of the important role of physical fitness for athletes' competitive performance, the General Administration of Sport of China issued the "Notice on Further Strengthening Basic Physical Training and Complementary Physical Ability" (hereinafter "the Notice") on February 24, 2020, which stipulates the competition system of combining physical fitness test with competition results. After that, the 2020 National Taekwondo Championship Series (NTCS) held in Wuxi, Jiangsu Province became the first taekwondo competition with PFT added to the rules, which also provided valuable data for the study of the relationship between athletes' competition performance and PFT.

The main aim of this paper is to study the relationship between PFT results and taekwondo athletes' competitive performance by analyzing the results of four competitions in the 2020 NTCS. As a supplement to the research on physiological indicators and PFT in training, this research hopes to use empirical research to provide a more macroscopic view of the role of physical fitness in Taekwondo competitions.

## 2. Materials and methods

The 2020 NTCS contained four competitions, two PFTs and a scoring system. The first and second competitions were held from September 22nd to 25th and September 27th to 30th, respectively. Prior to this, the first PFT (PFT1) was conducted from September 19th to 20th. The second PFT (PFT2), the third competition and the championship final (hereafter referred as the fourth competition) were held during October 15th to October 28th. Only athletes who have taken the PFT1 can participate in the first and second competitions, only athletes who have taken the PFT2 can participate in the third and fourth competitions.

A total of 638 professional Taekwondo athletes (393 males and 245 females) participated in the NTCS, all of whom were senior athletes from provincial teams in all provinces of China. All the athletes were required to compete in all four competitions and two PFTs. However, actually the number of athletes in each competition may vary slightly due to injuries or other reasons, e.g., school exams and unplanned restitution, etc. After each competition athletes were awarded points according to their competition ranking and the total number of points from the four competitions determines the final ranking of the athletes in the 2020 NTCS. The scoring rule of the NTCS was listed in **Table 1**. In summary, this could be calculated with the following formula: $S = [10 \times 166.7e^{-0.5108 \times M}]/10$, where S represents points, M is determined by the competition ranking and bracket means rounding function.

**Table 1. Scoring rule of the 2020 NTCS.**

| Rank | No. 1 | No. 2 | Joint third | Joint fifth |
|---|---|---|---|---|
| | (M = 1) | (M = 2) | (M = 3) | (M = 4) |
| Point | 100 | 60 | 36 | 21.6 |

The competition rules adopted for the NTCS is "World Taekwondo Competition Rules" (in force as of May 15, 2019) [21]. A three-round total points system was carried out for each match, each round is 2 minutes, and there is a 1-minute break between rounds. The PFT used in NTCS is similar to the German motor test [22], the US Army Air Forces PFT [23], the physical fitness scoring system for naval service personnel [24] and the national student fitness test in China [25]. Specifically, the 2020 NTCS PFT included five items, namely: weight-bearing squat, abdominal muscle endurance, back muscle endurance, 30-meter run and 3000-meter run. This battery of tests can be considered as an indicator stand for various aspects of an athlete's athletic ability, including agility, explosive power, strength, aerobic capacity and anaerobic capacity, which has been viewed as a multidimensional structure that reflects motor performance ability [26]. The grading standard is given in **Table 2**.

Weight-bearing squat is used to test the strength of the quadriceps, gluteus maximus and other lower limb muscles [27]. The evaluation standard of squat strength is the ratio of the maximum squat weight to the body weight. Athletes are required to stand with feet slightly wider than shoulders, and toes can be rotated 15–30 degrees externally, squat until the front of the thigh reaches or below the horizontal line, and then squat up. Each athlete will test 3 times

**Table 2. Assessment matrix for the physical fitness test.**

| Score | Weight-bearing squat (Multiples of weight) | 30-meter run (s) | Back muscle endurance (s) | Abdominal muscle endurance (s) | 3000-meter run (minute:second) | |
|---|---|---|---|---|---|---|
| | | | | | Male | Female |
| 20 | ≥1.2 | ≤4.5 | ≥120 | ≥120 | ≤11:00 | ≤11:30 |
| 19 | 1.18 | 4.55 | 118–119 | 118–119 | 11:01–11:10 | 11:31–11:40 |
| 18 | 1.16 | 4.6 | 116–117 | 116–117 | 11:11–11:20 | 11:41–11:50 |
| 17 | 1.14 | 4.65 | 114–115 | 114–115 | 11:21–11:25 | 11:51–11:55 |
| 16 | 1.12 | 4.7 | 112–113 | 112–113 | 11:26–11:30 | 11:56–12:00 |
| 15 | 1.1 | 4.75 | 110–111 | 110–111 | 11:31–11:35 | 12:01–12:05 |
| 14 | 1.08 | 4.8 | 108–109 | 108–109 | 11:36–11:40 | 12:06–12:10 |
| 13 | 1.06 | 4.85 | 106–107 | 106–107 | 11:41–11:45 | 12:11–12:15 |
| 12 | 1.04 | 4.9 | 104–105 | 104–105 | 11:46–11:50 | 12:16–12:20 |
| 11 | 1.02 | 4.95 | 102–103 | 102–103 | 11:51–11:55 | 12:21–12:25 |
| 10 | 1.0 | 5.0 | 100–101 | 100–101 | 11:56–12:00 | 12:26–12:30 |
| 9 | 0.95 | 5.05 | 95–99 | 95–99 | 12:01–12:10 | 12:31–12:40 |
| 8 | 0.9 | 5.1 | 90–94 | 90–94 | 12:11–12:20 | 12:41–12:50 |
| 7 | 0.85 | 5.15 | 85–89 | 85–89 | 12:21–12:30 | 12:51–13:00 |
| 6 | 0.8 | 5.2 | 80–84 | 80–84 | 12:31–12:45 | 13:01–13:15 |
| 5 | 0.75 | 5.25 | 75–79 | 75–79 | 12:46–13:00 | 13:16–13:30 |
| 4 | 0.7 | 5.3 | 70–74 | 70–74 | 13:01–13:15 | 13:31–13:45 |
| 3 | 0.65 | 5.35 | 65–69 | 65–69 | 13:16–13:30 | 13:46–14:00 |
| 2 | 0.6 | 5.4 | 60–64 | 60–64 | 13:31–13:45 | 14:01–14:15 |
| 1 | 0.55 | 5.45 | 30–59 | 30–59 | 13:46–14:00 | 14:16–14:30 |
| 0 | ≤0.5 | >5.5 | <30 | <30 | >14:00 | >14:30 |

and take the highest weight. To test the endurance of the abdominal muscles, the test subject lies supine on a jump box with the torso suspended [28], the anterior superior iliac spine on the edge of the box, the arms crossed over the chest and the lower legs held in place by a belt, keeping the body in one plane and recording the longest time the athlete can hold on. The test method of back muscle endurance is the same as abdominal muscle endurance, except that the athlete lies prone on the jumping box [28]. The 30-meter sprint test is designed to measure speed ability [22] and uses a standing start, each athlete takes the best score twice in the test, and pikes are not allowed in the test. The 3000 m run is a test of aerobic endurance [22] and each athlete will only be tested once. Both the 3000m timed run and the 30m sprint run were carried out on the 400-meter standard track and field field, and all the other equipment involved in the PFT and their models/parameters are given in the S1 Table. The PFT items, scoring criteria and test criteria are all set by the NTCS event organising committee, including the Chinese Taekwondo Association and the Jiangsu Provincial Sports Bureau. The organising committee invited five elite athletes (for ethical reasons they are named A1, A2, A3, A4 and A5) from the national team to take part in the PFT2, which on the one hand helps these elite athletes to understand what level of fitness they are at, and on the other hand helps other athletes to understand the difference between their own fitness and that of the elite athletes.

According to 2020 NTCS rules, athletes who score below 50 in the PFT are directly disqualified from the competition. After the top 8 athletes are determined in each competition, the top four of these eight athletes in terms of PFT ranking go directly to the semi-finals, while the last 4 athletes are regarded as tied for fifth. The two quarter-finalists who did not advance to the final are considered to be tied for third. Since the number of athletes in each weight category is different, the percentage of PFT ranking, defined as an athlete's PFT ranking divided by the number of athletes in his or her weight caterogy, is used in this research, regarding to it's more meaningful to facilitate the comparison among different weight categories.

In this research, data statistics, least squares fitting and linear regression analysis were all performed with MATLAB software version R2021a.

## 3. Results and discussion

### 3.1 Correlation between competition results and PFT results

**3.1.1 Total points.** **Fig 1** displays the proportion of athletes with different PFT ranking percentages among the athletes who have entered the top 8 and top 3 in the final points. The proportions were fitted to the results of PFT1, PFT2 and the average of these two ($P\bar{F}T$). It can be found that the proportions can be well fitted by exponential function with the shape of $y = ae^{bx}$, where $a$ and $b$ are constant coefficients. The coefficients calculated by the least squares method, the Goodness of Fit ($R^2$) and the number of athletes (N) are listed in **Tables 3** and **4**. For all athletes, female athletes and male athletes entering the top 8 (**Fig 1A–1C**), the $R^2$ are very encouraging and all above 0.8, although with a slight decline when fitted with PFT2 results. This result suggests that the final total points of both male and female athletes are highly positively correlated with their PFT scores, i.e. athletes with higher PFT scores are more likely to do better in the 2020 NTCS. The fitting in entering the top 3 seems even better, except for the fitting with the $P\bar{F}T$ of male athletes, which has the minimum $R^2$ value (0.788). Moreover, the fitting curves in **Fig 1D–1F** are steeper compared to **Fig 1A–1C**, meaning that generally only athletes ranked in the top 20% of the PFT are likely to be in the top 3 in the final total points.

Moreover, it is obvious that the exponential function can fit better than the linear function according to the distribution of data. This may be due to the fact that the PFT rank is the only basis for eight-to-four promotion, which means after entered the quarterfinals, the top 4

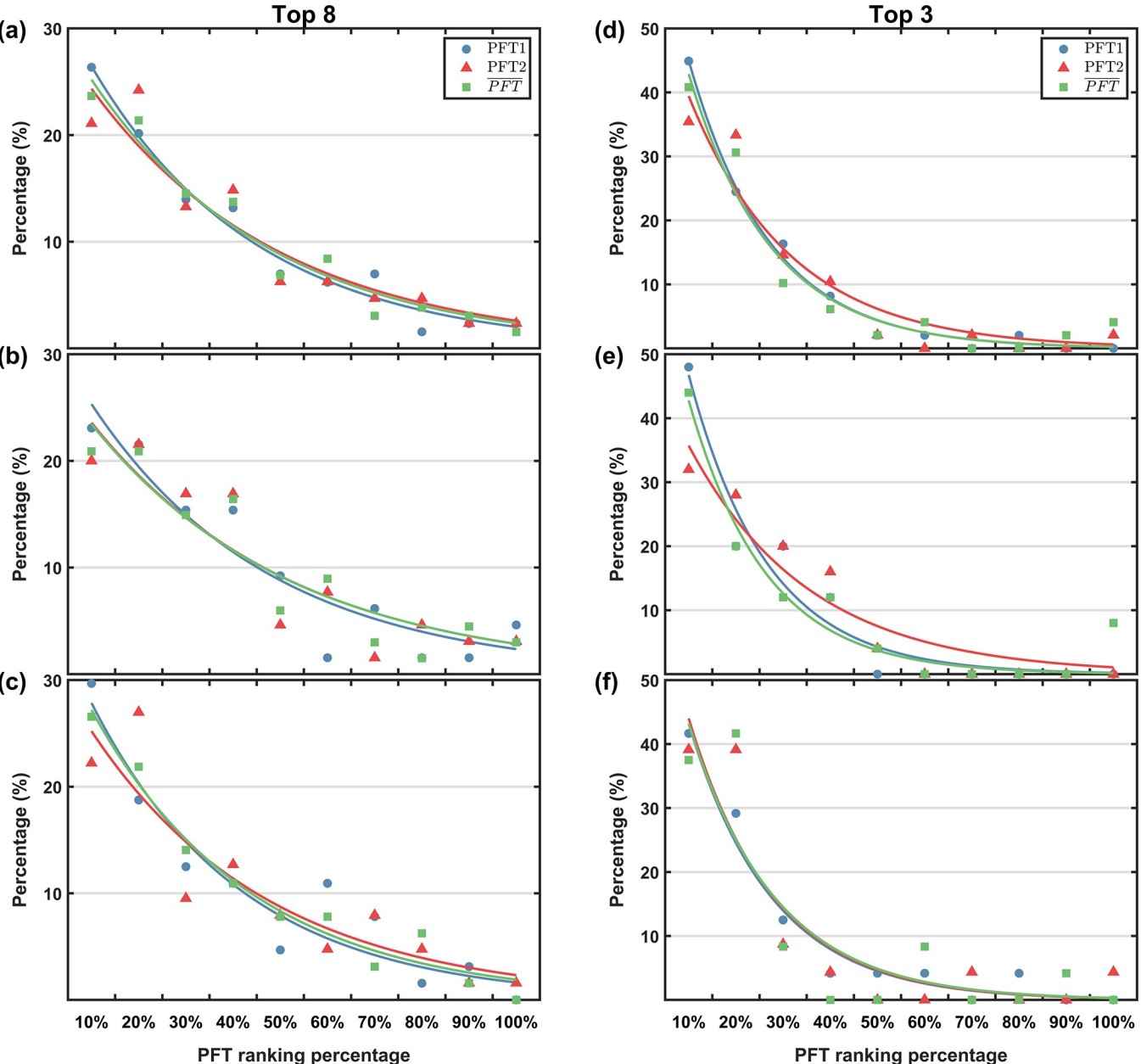

**Fig 1. Relationship between total points and physical test scores.** The percentage of all athletes (top row), female athletes (middle row) and male athletes (bottom row) in each PFT ranking segment among the top 8 (a-c) and top 3 (d-f) athletes in the final total points. The blue, red and green solid lines correspond to the fitting curves of PFT1, PFT2 and $\overline{PFT}$, respectively.

athletes in PFT will inevitably get more points than the bottom 4 ones. On the other hand, the exponential scoring rule also plays an important role in widening the points gap between athletes.

**3.1.2 Entering the quarterfinals.** As aforementioned, athletes who enter the semifinals are screened by PFT rankings. Thus, in order to study the relationship between athletes' physical fitness and competition performance, the impact of the rules must be excluded. In this section, the statistics are made on the PFT results of the top 8 athletes in each competition.

**Table 3. The fitting coefficient (a and b), the Goodness of Fit ($R^2$) and the number of athletes (N) in Fig 1A–1C.**

| | | a | b | $R^2$ | N |
|---|---|---|---|---|---|
| All | PFT1 | 0.305 ± 0.037 | -2.861 ± 0.505 | 0.974 | 573 |
| | PFT2 | 0.276 ± 0.067 | -2.490 ± 0.913 | 0.890 | 520 |
| | $\overline{PFT}$ | 0.287 ± 0.043 | -2.626 ± 0.579 | 0.959 | 638 |
| Female | PFT1 | 0.289 ± 0.072 | -2.641 ± 0.978 | 0.893 | 219 |
| | PFT2 | 0.265 ± 0.081 | -2.354 ± 1.097 | 0.834 | 206 |
| | $\overline{PFT}$ | 0.264 ± 0.068 | -2.346 ± 0.926 | 0.873 | 245 |
| Male | PFT1 | 0.326 ± 0.079 | -3.149 ± 1.093 | 0.907 | 354 |
| | PFT2 | 0.288 ± 0.094 | -2.649 ± 1.280 | 0.827 | 314 |
| | $\overline{PFT}$ | 0.315 ± 0.043 | -2.956 ± 0.591 | 0.968 | 393 |

[a]The coefficients a and b are in the form of mean and 99% confidence interval.

Fig 2 illustrates that, in general, the top 10% of athletes in PFT ranking account for the highest proportion of the top 8 athletes in the four competitions. As the PFT ranks lower, this proportion is also lowers. This feature is more pronounced for the statistics of all athletes (Fig 2A–2D) than for females (Fig 2E–2H) and males (Fig 2I–2L) specifically. In addition, the proportion of athletes in the top 50% and bottom 50% of the PFT was also counted, which shown in Fig 2 with red ladders. Apparently, the top 50% account for a higher proportion of the top 8 athletes (about 2/3). For female Taekwondo athletes, the top 50% in the PFT accounted for a higher proportion in the top 8 (Fig 2E–2H), implying that compared with male athletes, physical fitness is more crucial in competition performance for female athletes. A linear correlation analysis was performed on the proportions of athletes with different PFT ranking percentages among all the top 8 athletes (Fig 3), all coefficients are calculated within the 95% confidence interval and p value less than 0.001. The high correlation coefficient (r = 0.914) illustrates that there is a strong negative correlation between an athlete's probability of achieving a top 8 ranking in a competition and their percentage ranking in the PFT. Given the sample size of athletes in this study is 638, the results are considered to be statistically significant. However, it also should be noted that due to the lack of data on all athletes' years of training and whether they have achieved international acclaims, we cannot exclude the effect of these covariates on athletes' competitive performance.

**Table 4. The fitting coefficient (a and b), the Goodness of Fit ($R^2$) and the number of athletes (N) in Fig 1D and 1E.**

| | | a | b | $R^2$ |
|---|---|---|---|---|
| All | PFT1 | 0.602 ± 0.054 | -5.811 ± 0.719 | 0.992 |
| | PFT2 | 0.497 ± 0.135 | -4.635 ± 1.753 | 0.925 |
| | $\overline{PFT}$ | 0.570 ± 0.129 | -5.710 ± 1.771 | 0.951 |
| Female | PFT1 | 0.630 ± 0.151 | -5.975 ± 1.958 | 0.945 |
| | PFT2 | 0.434 ± 0.120 | -3.898 ± 1.520 | 0.916 |
| | $\overline{PFT}$ | 0.581 ± 0.149 | -6.104 ± 2.137 | 0.938 |
| Male | PFT1 | 0.573 ± 0.102 | -5.641 ± 1.386 | 0.969 |
| | PFT2 | 0.582 ± 0.243 | -5.600 ± 3.218 | 0.854 |
| | $\overline{PFT}$ | 0.567 ± 0.295 | -5.453 ± 3.899 | 0.788 |

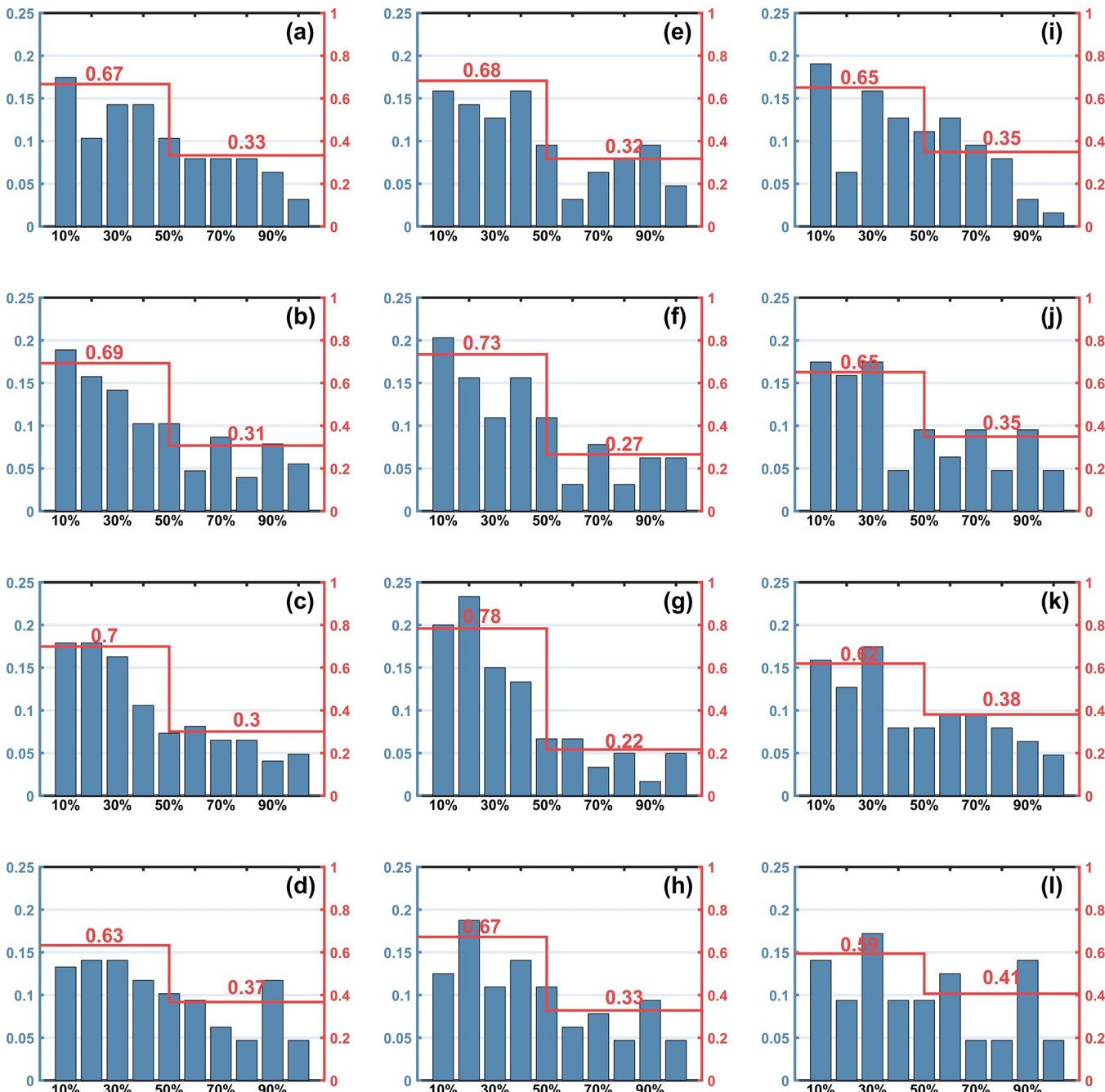

**Fig 2. Percentage of athletes of different PFT rankings who entering the quarterfinals.** The percentage of all athletes (a-d), female athletes (e-h) and male athletes (i-l) ranked by PFT among the top 8 athletes in the four competitions, each row in this figure is the results of one competition. The red ladders and numbers indicate the percentage of athletes in the top 50% and bottom 50% of the PFT who entering the quarterfinals.

**3.1.3 Winning in semifinals.** This section will discuss whether athletes with better physical fitness are more likely to win among the 4 athletes who have entered the semifinals. **Table 5** counted the frequencies of the first, second, third and fourth PFT ranked athletes of the four athletes who have entered the semifinals won the first place. Since both female and male competitors are divided into eight weight categories, there are a total of 16 semifinals in

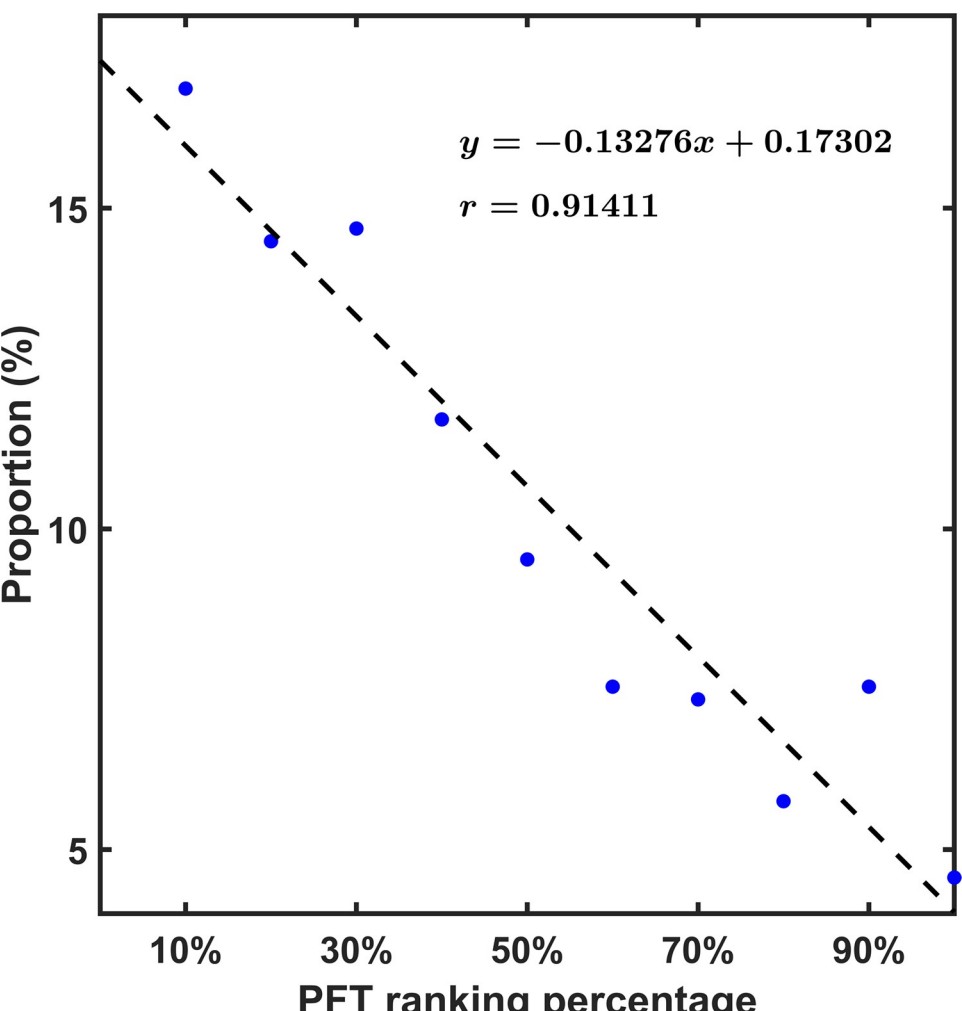

**Fig 3. Linear regression of the proportion of athletes in each PFT ranking interval among all the top 8 athletes in four competitions.** The regression equation and correlation coefficient are also listed.

each competition, i.e., the sum of the numbers in each row of Table 5 is 16. For example, the number '4' in the first row and first column means that in the 8 female's semifinals of the first competition, 4 of the athletes who ranked first in the PFT won the championship. Clearly, assuming that every athlete who enters the semifinals has an equal chance of winning the first place, the expectation of this value should be 2.

**Table 5. Frequency statistics of top 4 athletes with different PFT rankings winning the first place.**

| PFT rank | No. 1 | | No. 2 | | No. 3 | | No. 4 | |
|---|---|---|---|---|---|---|---|---|
| Competition | Female | Male | Female | Male | Female | Male | Female | Male |
| 1 | 4 | 3 | 1 | 3 | 2 | 2 | 1 | 0 |
| 2 | 3 | 3 | 3 | 2 | 1 | 1 | 1 | 2 |
| 3 | 4 | 3 | 2 | 2 | 1 | 2 | 1 | 1 |
| 4 | 5 | 3 | 0 | 4 | 2 | 1 | 1 | 0 |
| Average | 4 | 3 | 1.5 | 2.75 | 1.5 | 1.5 | 1 | 0.75 |

[a]The PFT ranking is only a relative ranking of the four athletes who have entered the quarterfinals.

For females, the average number of times that the athlete ranked first in the PFT among those entering semifinals won the championship is 4, which is twice the expected value. The second and the third ranked athletes in the PFT have the same frequency of winning the championship, both are 1.5, which is slightly less than the expected value. The athletes ranked last in the PFT have the least frequency of winning. The same is true for male athletes, but the difference in the frequency of winning championship between the first and the second in the PFT is not as large as that of female athletes. However, the above results all confirm that PFT results are indeed related to athletes' competitive performance, that is, athletes with better physical fitness have a greater probability of having better competitive performance. Furthermore, for female athletes entering the semifinals, the advantage brought by physical fitness may be even greater, this result is consistent with the findings of previous studies [18].

## 3.2 Key physical indicators

The scores of the athletes' 5 individual PFT items are respectively counted (**Fig 4**). As is illustrated, almost all athletes scored 19–20 points in weight-bearing squat, abdominal muscle endurance and back muscle endurance. This means that the squat strength, abdominal muscle endurance and back muscle endurance of these professional Taekwondo athletes all achieved the standard of excellence set by the General Administration of Sport of China. The gap between athletes' PFT results mostly comes from the 30-meter sprint. Female athletes generally did not score high in the 30-meter sprint (**Fig 4D**), almost all of them scored less than 10 points. Nevertheless, compared with other athletes, the female athletes who have entered the top 8 in the competition has relatively higher scores. Male athletes had higher scores than the female athletes in the 30-meter sprint (**Fig 4I**), and similarly, top 8 athletes score higher than other athletes. Male athletes also generally perform better than female athletes in the 3000-meter run, and compared with other athletes, the percentage of top 8 athletes getting high scores is only slightly larger. For female athletes, the top 8 athletes also have a slightly better 3000m performance than other athletes.

Therefore, athletes with better competition results in the 2020 NTCS (those who have entered the top 8 in the competitions) tended to have higher 30-m sprint socres than other athletes demonstrates that the physical fitness gap of Taekwondo athletes mainly depends on speed ability and explosive power, which is similar to the results of previous studies [12, 29, 30]. This is also consistent with the characteristics that taekwondo is an intermittent high-intensity sport. Therefore, we can easily recommend that female athletes should pay more attention to improving their physical fitness. Especially after entered the semifinals, due to intensifying competition, the advantage of physical fitness will be more obvious.

For professional athletes, there is not much difference in leg strength, abdominal muscles, back muscle endurance and aerobic capacity, but apart from the fact that the athletes all had higher levels of these fitness categories, another potential reason could be that the scoring criteria for weighted squats, abdominal muscular endurance and back muscular endurance were too low to clearly differentiate between the athletes' fitness levels in these categories. The low variability of athlete scores in these three items limits the discussion of the impact of these fitness indicators on the competitive performance of taekwondo athletes and is one of the limitations of this study.

## 3.3 PFT total scores

The average and standard deviation of PFT total scores of different weight categories athletes are counted in this section. **Fig 5** shows that the PFT scores of female athletes are generally lower than that of male athletes, although the PFT standards of female and male are different.

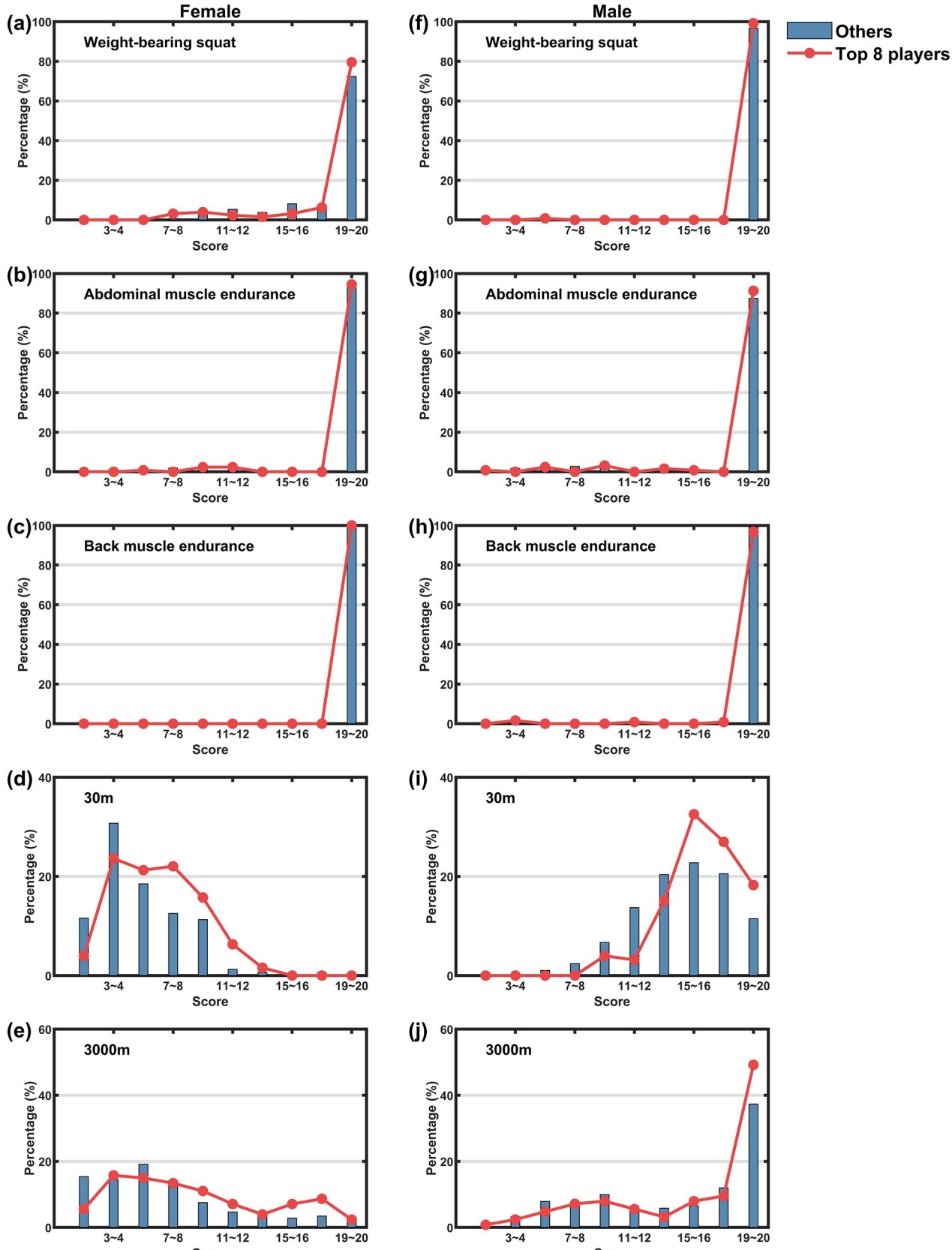

**Fig 4. The proportion of top 8 athletes and other athletes in each score segment of each individual PFT item.** The left column is for female athletes, and the right column is for male athletes.

For both female and male athletes, the higher weight category athletes tend to obtain lower PFT scores. According to results mentioned in the previous section, this could be due to athletes with higher weights typically having relatively weaker explosive power than those with smaller weights.

Another interesting thing is that for almost every weight category class, the average score of the PFT2 is better than that of the PFT1. This may explain the results in section 3.1.1. That is, after the PFT1, the first and the second competitions, the athletes realized that their physical deficiencies affected their performance in the competition, so they trained to improve their physical fitness. And with its improvement, the physical fitness gap between competitors has less impact on the competition, that's considered to be a reasonable explanation for the slight decline in $R^2$ when fitting with PFT2 ranking percentages.

In addition, thanks to the support of the Chinese National Taekwondo Team, several Taekwondo athletes who had won Olympic or World Championships also participated in the second PFT2, and their PFT scores are also marked in **Fig 5**. These former world champion athletes all had fitness test results above the average of the other athletes, particularly scores in A1, A2 and A3, which exceeded the average plus one standard deviation. This result also provides a strong evidence that physical fitness is a necessary condition for excellent Taekwondo athletes.

## 4. Conclusions

This study conducted a statistical analysis of the competition results and PFT results of the 2020 NTCS, which is the first Taekwondo major event after the General Administration of Sport of China issued the Notice and provided a relatively large data set. Overall, the results

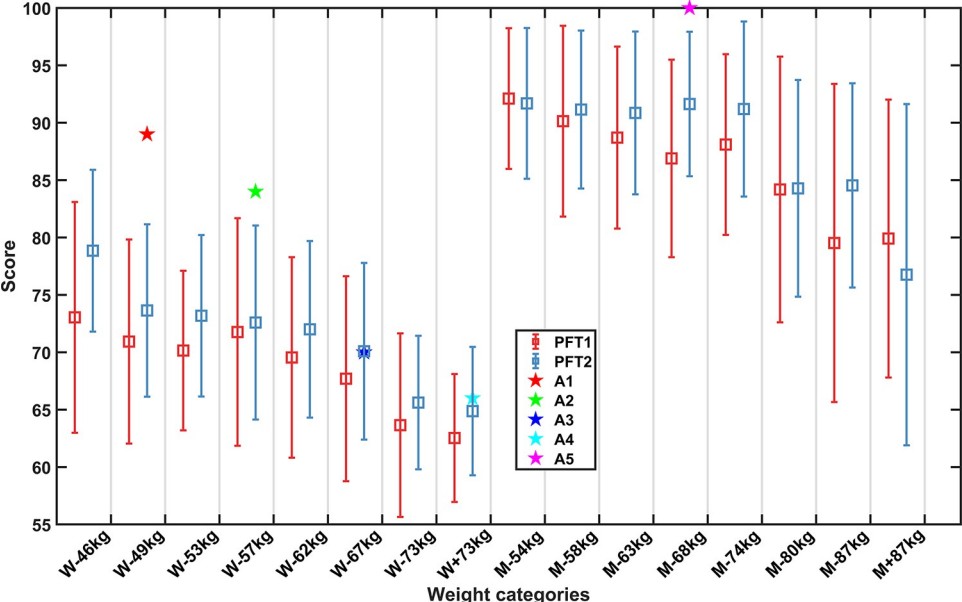

**Fig 5. PFT scores of athletes in different weight categories.** Mean value and standard deviation are represented by box and solid line respectively (p < 0.001). Scores of the national team athletes are marked by five-pointed stars.

demonstrate there is a significant positive correlation between athletes' competition performance and PFT ranking.

As athletes rank lower in PFT, their proportion of the top 8 athletes in the final total points decreases exponentially, and the proportion decline even faster of the top 3. In each competition, the top 50% in PFT account for more in the athletes entering quarterfinal. And in the semi-finals, athletes who are also ranked first in the PFT have a greater chance of coming to win the championship. The above three results all imply that the difference in physical fitness does play a key role in the athletes' competition performance. For female athletes, the difference in physical fitness is relatively large, and there is also a large room for improvement in physical fitness; this conclusion is consistent with the conclusions in other related studies [12, 18]. Especially, the key to the difference of Taekwondo Athletes' physical fitness lies in their explosive power, which is reflected in the biggest difference in the score of 30 meter sprint.

Furthermore, due to the PFT scores are associated with qualification for promotion in the 2020 NTCS, the athletes attached great importance to it, which makes this research more meaningful. This study is also a supplement to research on physiological indicators, as it analyses and provides a macroscopic view of the role of physical fitness in Taekwondo competitions.

## Supporting information

**S1 Table. The information about the equipments used in the PFT.**
(XLSX)

**S1 Data. Raw data.**
(XLSX)

## Acknowledgments

The authors wish to thank all the athletes participated in the 2020 National Taekwondo Championship Series.

## Author Contributions

**Data curation:** Lumin He.

**Investigation:** Lumin He.

**Methodology:** Rui Liu.

**Software:** Rui Liu.

**Supervision:** Lumin He.

**Visualization:** Rui Liu.

**Writing – original draft:** Rui Liu.

**Writing – review & editing:** Lumin He.

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
