## [Decision Letter · Decision Letter 0]

21 Jul 2021

PONE-D-21-18355

The relationship between physical fitness and competitive performance of Taekwondo athletes

PLOS ONE

Dear Dr. He,

Thank you for submitting your manuscript to PLOS ONE. After careful consideration, we feel that it has merit but does not fully meet PLOS ONE’s publication criteria as it currently stands. Therefore, we invite you to submit a revised version of the manuscript that addresses the points raised during the review process.

We look forward to receiving your revised manuscript.

Kind regards,

Krzysztof Durkalec-Michalski, Ph.D

Academic Editor

PLOS ONE

Journal Requirements:

Reviewers' comments:

Reviewer's Responses to Questions

**Comments to the Author**

1. Is the manuscript technically sound, and do the data support the conclusions?

Reviewer #1: Partly

Reviewer #2: Partly

Reviewer #3: Partly

2. Has the statistical analysis been performed appropriately and rigorously? 

Reviewer #1: Yes

Reviewer #2: I Don't Know

Reviewer #3: Yes

3. Have the authors made all data underlying the findings in their manuscript fully available?

Reviewer #1: Yes

Reviewer #2: No

Reviewer #3: Yes

4. Is the manuscript presented in an intelligible fashion and written in standard English?

Reviewer #1: Yes

Reviewer #2: No

Reviewer #3: No

5. Review Comments to the Author

Reviewer #1: First and foremost, I want to congratulate the research team for undertaking an extremely difficult task, which without a shadow of doubt is linking physical fitness tests (PFT) with the actual competitive performance of professional athletes. What is more, I had little problems to understand the text in overall, despite the fact that every now and then, there were some mishaps (for instance line 190, were an unwanted “n” appeared).

I believe that martial arts practitioners (athletes and coaches alike) will find this study interesting and noteworthy.

However, in my humble opinion, there are some key factors and issues to address, in order to allow the article to be deemed worthy for publication. I shall list all of the doubts and remarks below, each in an corresponding section.

Introduction:

The cited publications do correspond with the article’s main narrative, and even in some cases directly on point. However, especially in recent years, there were some noteworthy publications regarding competitive performance, which would definitely enrich this particular section and prove the author’s scientific prowess. Please bear in mind, that any hope of tackling a macroscopic view approach, should be reflected in a slightly broadened analysis of recent scientific findings.

Materials and methods:

Starting of with a praise, the sheer amount of athletes measured is quite impressing. Inclusion of some of the most proficient taekwondo practitioners is a positive definitely worth mentioning. Another strong point is the use of MATLAB software, which is a modern tool for analysis. Unfortunately, there are some doubts on my behalf:

1) The description of the PFT test corresponds with the actual procedure of the said test, still there is no indication of any citation with scientific recognition purely devoted towards description of said test. In order to counter any doubts about the choice of PFT, appropriate citations seems more than necessary.

2) What kind of equipment was used to determine the outcomes of each of the 5 items in the PFT test? Please attach all of the manufacturer’s details, version/batch number, software release, etc. Even the most trivial items used in the study (i.e. jumping box, stopwatch), without which the test cannot be recreated otherwise, need to be described.

3) Addressing the phrases from lines 85-86, it seems only viable, to at least name the reasons (just this once) of exclusions in a bracket, i.e. (military duties, school exams, unplanned restitution, etc.).

4) After careful analysis of the attached citations, it would seem that the most troublesome aspect of the whole study is. Why did the authors decided not to include any type of a bio-monitor throughout the whole course of the research? Most of the research teams of the cited articles, did use a wide variety of scientific equipment to precisely assess the fitness level of their subjects. This is usually done in connection with main study method (like PFT in this study) in order to prove the eligibility and accuracy of the method itself. As a side note Bio-monitors record indices such as time, speed, body temperature, physiological load, mechanical load, etc. What is more, they were allowed for use, by the World Taekwondo officials during tournaments.

Results and discussion:

There was some but little struggle with understanding and following the narrative. The presented results, as well as tables, were clear to fathom upon careful inspection. Still due to sheer number of data, it was slightly difficult to decipher and process everything as a whole. I wonder if this could be remedied in any way? However, not at the cost of the data’s quality of display. If anyone would take all of the reasoning and explanations indiscriminately, then there would be no more remarks whatsoever. Unfortunately that is not the case and the most troublesome issues include:

5) Why did the research team opt not to carry out an analysis of covariance (as “main course” or as an addendum)? As stated earlier, the amount of research participants is impressive and would suffice for said analysis. It would seem beneficial to apply covariates such as:

a. Seniority (junior/senior)

b. International acclaim

c. Years of combat experience (truly not the same as point b)

d. Day of competition/testing (as stated in the article itself, there were multiple measure periods)

e. Other factors that the research team deems worthy of adding and are aware of

6) As far as I am concerned, the authors should specifically underline that only one tournament was monitored and continuing the narrative from remark number nr.5, challenge themselves against the knowledge from articles they have cited in this paper, how this can be seen as a limitation. Please bear in mind, that in at least of three of the citations, scientific data was gathered from more than one tournament.

7) Multiple times did the authors underline the benefits of introducing the PFT testing as viable method to predict and assess competitive performance of athletes. I find the presented results minimally too broad. If possible, could the authors try to make the discussion even more interesting, by speculating how should thresholds or ranges thereof look like for different types of performance. Putting it simple, how much PFT score is necessary for qualifying to cup stage, quarter finals, semis etc.?

Conclusions:

Please apply sufficient changes in the conclusions, after careful considerations of the remarks stated above. If point number 7 does seem viable for the authors, it most definitely should be sounded out in this section specifically.

I truly await responses for the above mentioned insights.

Reviewer #2: The study presents the results of original research. Described research is very interesting and conducting them during the competition time was the innovative approach. Definitely, the change of scoring rules in taekwondo has influenced the physical characteristics of top athletes. Such research can help create and develop the training programs for taekwondo competitors.

Introduction

line 39 I’m not sure what is meant by “increase in the antagonism between two competitors”. The whole sentence could be deleted.

line 40 “previous rules” maybe giving a year when they were introduce will guide the reader which rules were meant

Methods

line 77 chart of a timeline would increase the readability of the process. What is the first and second race? Do you mean the subsequent matches leading to the finals?

Did the same athletes take part in consequent races or is that organised for different groups of athletes? The number of participants suggested the second way but it was not described clearly.

Were the PFT results taken into account to be included in the further stage of competition or it didn’t have an influence on the result of the competition? It is crucial to understand how the both scores relate with each other.

line 80 change “fourth races” to “fourth race”

line 82 I believe that authors refer to Taekwondo Competition Rules, not PARA Taekwondo Competition Rules. It needs to be explained or changed.

line 84 what is meant by “professional” - does it mean that the competition was closed to amatours? If that’s true, adding selection criteria for competition would help to understand the selection process.

Other participants' characteristics are missing. No information re the age range. Is the National Taekwondo Championship Series for seniors only or junior athletes are also competing?

In the results the differentiation between the national team was addressed - it should be described in the methods section.

line 96 Table 1. How has the grading standard developed? Any references? Why is only one of the PFT grading standards different for female and male athletes?

line 99 word “produced” should be changed to “determined” or similar

Ethics statement should be provided for this study, although authors state that it is not required. It is Human Subject Research involving human participants.

Statistical methods should be described. Why and which analysis were conducted? No explanation can be found in this section and it’s difficult to find the clarity in the results descriptions.

Although authors suggest full availability of the data, they are available on the website of Chinese Taekwondo Association in Chinese. It is stated that “if anyone needs it in English, feel free to contact the corresponding author” but the english version was not provided to the reviewer. The accessibility is somehow restricted and cannot be evaluated by English-speaking reviewer.

Results

This part of the manuscript is divided into sections - the same structure could be adapted to describe statistical analysis in the Methods section.

line 107 Total points calculation should be moved to the methods section

line 133-136 The title and description of the figure should be separated.

line 149 “quarterfinals” how this term relates to the term race? The descriptions in the methods section should clearly explain that.

line 183 “eight heavyweights'' should be changed into “weight categories” or “weight divisions”. Although standard English is used, some terms in the text are misleading.

line 253 “quite good” should be changed to different, more objective term

line 255 and Fig 5 names of participants were presented - did the researchers receive the consent to use the personal details? Prior to sharing human research participant data, authors should consult with an ethics committee to ensure data are shared in accordance with participant consent and all applicable local laws. Data sharing should never compromise participant privacy. It is therefore not appropriate to publicly share personally identifiable data on human research participants.

Knowing the rationale for the inclusion of each analysis and having a clear description of which participants, from which stage of competition took part in it, would have increased the understanding of the whole research design. I believe that each analysis was conducted with the purpose but neither methodological section nor the discussion explains their value and intention in an intelligible fashion.

Conclusions

Part of this section should be added to the discussion part of the manuscript.

Had the research questions been clearly and specifically formulated in the introduction or methods, the conclusions could have been more precise.

Reviewer #3: Thank you for your interesting paper.

There are several concerns that you need to address before your paper would be publishable. Please look at my comment on the pdf version of your paper.

In summary, you need to used proper terminology with regards to Taekwondo. For example we do not use “race” in TKD but “match” or competition. there are other vocabulary that I highlighted for you.

You need to include the studies on the validity and reliability of the Physical Fitness Tests (PFT) used in this study. Who devised The scoring system used in your study and was it tested and reliable?

Your tables are very confusing and need to be redone. You have used too many tables and figures. you need to decrease the numbers and do not use table and figure to state the same data.

Your discussion portion of your paper lacks the comparison with other studies and stating the reasons behind your finding in light of the literature. This is very crucial and essential for the success of a paper.

I would not be able to agree with your conclusion since you have not provided evidence for the reliability and validity of the tests, scoring systems etc.

One last but most concerting is the ethical issues of this study. It is unethical to include the name of athletes or any identifying data about the athlete in a scientific peer reviewed paper.

6. PLOS authors have the option to publish the peer review history of their article (what does this mean?). If published, this will include your full peer review and any attached files.

Reviewer #1: **Yes: **Doctor Engineer Michał Janowski

Reviewer #2: No

Reviewer #3: No

---

## [Author Response · Author response to Decision Letter 0]

17 Jan 2022

List of Responses

Dear Editors and Reviewers:

Thank you for your letter and for the reviewers’ comments concerning our manuscript entitled “The relationship between physical fitness and competitive performance of Taekwondo athletes” (ID: PONE-D-21-18355). Those comments are all valuable and very helpful for revising and improving our paper, and provides guidance to us. We have studied the comments carefully and made revisions accordingly. In addition, we also upload the raw data used in our research (in supporting information), which is translated and summarised from the data in Chinese. The raw data in Chinese can be downloaded from the Chinese Taekwondo Association website (http://www.taekwondo.org.cn/). The main changes in the paper and responses to reviewers’ comments are as follows:

Reviewer #1: 

1) The description of the PFT test corresponds with the actual procedure of the said test, still there is no indication of any citation with scientific recognition purely devoted towards description of said test. In order to counter any doubts about the choice of PFT, appropriate citations seems more than necessary.

Response: Thanks for your suggestion, now we have added some literatures to support the PFT used in the NTCS that we invesgated in our study (Line 101-103 in the revised manuscript).

2) What kind of equipment was used to determine the outcomes of each of the 5 items in the PFT test? Please attach all of the manufacturer’s details, version/batch number, software release, etc. Even the most trivial items used in the study (i.e. jumping box, stopwatch), without which the test cannot be recreated otherwise, need to be described.

Response: The information about the equipments are listed in the Supporting information now, including the length, weight and diameter of the barbell bar, the weight of barbell, the weight of barbell clamp, the parameter of jumping box, and the model of stopwatch. However, we don’t have access to the more detailed information, such as the manufacturers of barbell and jumping box. 

3) Addressing the phrases from lines 85-86, it seems only viable, to at least name the reasons (just this once) of exclusions in a bracket, i.e. (military duties, school exams, unplanned restitution, etc.).

Response: Thanks for your reminder, we have added a few more specific reasons to explain that (Line 88 in the revised manuscript). 

4) After careful analysis of the attached citations, it would seem that the most troublesome aspect of the whole study is. Why did the authors decided not to include any type of a bio-monitor throughout the whole course of the research?

Most of the research teams of the cited articles, did use a wide variety of scientific equipment to precisely assess the fitness level of their subjects. This is usually done in connection with main study method (like PFT in this study) in order to prove the eligibility and accuracy of the method itself. As a side note Bio-monitors record indices such as time, speed, body temperature, physiological load, mechanical load, etc. What is more, they were allowed for use, by the World Taekwondo officials during tournaments.

Response: Thanks for your insight into this issue. Indeed, the relationship between physiological load, mechanical load and athlete performance has been studied extensively, but our aim in this paper is to investigate the relationship between the PFT test scores and the athletic performance of taekwondo athletes. To the best of our knowledge, similar PFTs are used in the German motor test, the US Army Air Forces test, the physical fitness scoring system for naval service personnel and the national student fitness test in China (Line 101-103 in the revised manuscript). These fitness tests are quicker and more efficient than biological tests and, although more crude, reflect the macro-athletic ability of the athlete. Another equally important reason is that we only have access to data from PFT tests and not biomonitoring data, although they may exist. 

5) Why did the research team opt not to carry out an analysis of covariance (as “main course” or as an addendum)? As stated earlier, the amount of research participants is impressive and would suffice for said analysis. It would seem beneficial to apply covariates such as:

a. Seniority (junior/senior)

b. International acclaim

c. Years of combat experience (truly not the same as point b)

d. Day of competition/testing (as stated in the article itself, there were multiple measure periods)

e. Other factors that the research team deems worthy of adding and are aware of

Response: Thanks for your valuable comments and suggestions. Analysis of covariance is indeed a powerful tool for excluding the influence of uncontrollable and uninteresting variables on the results, however, we do not seem to be in a position to do this analysis for several reasons. The first and most important being that we do not have relevant data such as the number of years of training of the athletes, the specific days on which each athlete took the physical tests (for example, we only know that the first phase of the PFT took place on the two days of 19 September and 20 September). Regarding the several covariates you mentioned that may have an impact on the results, 

a. Seniority (junior/senior): According to the rules of the NTCS, the participants of the 2020 NTCS are professional taekwondo athletes, from the provincial or municipal teams of each province, not amateur athletes. If classify them as junior and senior, they would all be senior athletes.

b. International acclaim: We have no way of knowing whether they have received international acclaim or not.

c. Years of combat experience: As we explained earlier, this data is not available to us.

d. Day of competition/testing: The physical tests and the exact dates of each competition are undisclosed (only time slots are disclosed, for example the first leg starts on 22 September and ends on 25 September). On the other hand, for the single PFT, which lasted two days, this time difference was actually considered negligible, whereas for the two PFTs, the difference in performance due to the adjustment of the athletes' training plan was much greater. As analysed in the manuscript, after the first PFT the athletes discovered that the 30 m sprint was the key to the difference in ranks, and as a result the average grade in this item improved significantly in the second PFT.

We believe that in addition to PFT results, these variables you mentioned also have an impact on athletes' performance in competition, and the lack of such data is a major limitation of this paper, so we have included that section in the final conclusions section (Line 313-316 in the revised version).

6) As far as I am concerned, the authors should specifically underline that only one tournament was monitored and continuing the narrative from remark number nr.5, challenge themselves against the knowledge from articles they have cited in this paper, how this can be seen as a limitation. Please bear in mind, that in at least of three of the citations, scientific data was gathered from more than one tournament.

Response: Thanks for your valuable comments. However, we do not consider this to be a limitation of our study, in the literatures we cited, some collected data from multiple matches (e.g., Janowski et al., 2020; Sadowski et al., 2012; Kim et al., 2019), while others indeed collected data from only one match (e.g., Maloney et al., 2018; Bridge et al., 2014; Miller et al., 2011). For the 2020 NTCS we are studying, it is a series of four competitions, each of which is complete, from the preliminaries to the final, so it’s actually not just one competition (Line 76-83 in the revised manuscript).

Reference:

1. Janowski M, Zieliński J, Ciekot-Sołtysiak M, Schneider A, Kusy K. The effect of sports rules amendments on exercise intensity during taekwondo-specific workouts. Int J Environ Res Publ Health, 2020; 17(18): 6779.

2. Sadowski J, Gierczuk D, Miller J, Cieslinski I, Buszta M. Success factors in male WTF taekwondo juniors. J Combat Sports Martial Arts. 2012;1, 47-51.

3. Kim H-B, Jung H-C, Song J-K, Chai J-H, Lee E-J. A follow-up study on the physique, body composition, physical fitness, and isokinetic strength of female collegiate Taekwondo athletes. J Exerc Rehabil. 2015;11(1): 57.

4. Maloney MA, Renshaw I, Headrick J, Martin DT, Farrow D. Taekwondo fighting in training does not simulate the affective and cognitive demands of competition: Implications for behavior and transfer. Front Psychol. 2018;9(JAN): 25.

5. Bridge CA, Ferreira Da Silva Santos J, Chaabène H, Pieter W, Franchini E. Physical and physiological profiles of taekwondo athletes[J]. Sports Medicine. 2014;44(6): 713-733.

6. Miller J, Bujak Z, Miller M. Sports result vs. general physical fitness level of junior taekwondo athletes. J Combat Sport Martial Arts. 2011;2(1): 39-44.

7) Multiple times did the authors underline the benefits of introducing the PFT testing as viable method to predict and assess competitive performance of athletes. I find the presented results minimally too broad. If possible, could the authors try to make the discussion even more interesting, by speculating how should thresholds or ranges thereof look like for different types of performance. Putting it simple, how much PFT score is necessary for qualifying to cup stage, quarter finals, semis etc.?

Response: Thank you for your suggestion that we should enrich the discussion, and thanks for the hint. In fact, it’s hard to give a clear threshold because the fitness level of the players varies from weight categories. We have discussed a similar issue as much as possible (section 3.1.2), the results illustrate that there is a significantly negative correlation between the PFT ranking percentage and its proportion in the top 8 athletes, for example, an athlete with a PFT ranking of 10% or less has a 17% chance of reaching the top eight in the competition, while an athlete with a PFT ranking of 70%-80% has only about a 6% chance of reaching the top eight (as shown in Fig 3).

 

Reviewer #2:

Introduction

- line 39 I’m not sure what is meant by “increase in the antagonism between two competitors”. The whole sentence could be deleted.

Response: This statement was indeed somewhat subjective and unsupported by the literature, and we have removed it at your suggestion.

- line 40 “previous rules” maybe giving a year when they were introduce will guide the reader which rules were meant

Response: Thanks for your insight into this, we have modified it to “pre-2015 rules” (Line 39 in the revised manuscript).

Methods

- line 77 chart of a timeline would increase the readability of the process. What is the first and second race? Do you mean the subsequent matches leading to the finals?

- Did the same athletes take part in consequent races or is that organised for different groups of athletes? The number of participants suggested the second way but it was not described clearly.

- Were the PFT results taken into account to be included in the further stage of competition or it didn’t have an influence on the result of the competition? It is crucial to understand how the both scores relate with each other.

Response: We apologise for the unclear and misleading statement about the process. We have reorganised this part of the description (Section 2. Materials and methods). 

 Specifically, the 2020 NTCS contains four competitions and used a scoring system, (Line 76 in the revised manuscript), all the athletes were required to compete in all four competitions (Line 86-87). After each competition athletes were awarded points according to their competition ranking and the total number of points from the four competitions determines the final ranking of the athletes in the 2020 NTCS.

According to 2020 NTCS rules, athletes who score below 50 in the PFT are directly disqualified from the competition. After the top 8 players are determined in each competition in each competition, the top four of these eight athletes in terms of PFT ranking go directly to the semi-finals, while the last 4 players are regarded as tied for fifth.

- line 80 change “fourth races” to “fourth race”

- line 82 I believe that authors refer to Taekwondo Competition Rules, not PARA Taekwondo Competition Rules. It needs to be explained or changed.

Response: Thanks for your insight into this, we have modified these accordingly.

- line 84 what is meant by “professional” - does it mean that the competition was closed to amatours? If that’s true, adding selection criteria for competition would help to understand the selection process.

- Other participants' characteristics are missing. No information re the age range. Is the National Taekwondo Championship Series for seniors only or junior athletes are also competing?

Response: Yes, it means the 2020 NTCS was closed to amatours, and all the athletes were senior athletes from provincial teams in all provinces of China (Line 85-86). Unfortunately, however, the organising committee does not disclose data on the age of these athletes, their years of training and whether they have won international acclaim. This is a limitation of our study and we have added this part of discription in the Line 313-316.

- In the results the differentiation between the national team was addressed - it should be described in the methods section.

Response: Thanks for your advice, it is now added in the Line 124-128.

- line 96 Table 1. How has the grading standard developed? Any references? Why is only one of the PFT grading standards different for female and male athletes?

Response: The PFT items, scoring criteria and test criteria are all decided by the NTCS event organising committee, including the Chinese Taekwondo Association and the Jiangsu Provincial Sports Bureau. (We have added this sentence to the revised manuscript, in Line 122-124). For the scoring criteria used in the 2020 NTCS PFT, we found no published literature to prove its reliability. On the other hand, as we used the rankings of the athletes' fitness tests in our analysis, we avoided, as far as possible, the bias caused by possible uncertainties in the scoring system.

- line 99 word “produced” should be changed to “determined” or similar

Response: This word has been replaced.

- Ethics statement should be provided for this study, although authors state that it is not required. It is Human Subject Research involving human participants.

Response: We have added the Ethics statement in Line 331-335.

- Statistical methods should be described. Why and which analysis were conducted? No explanation can be found in this section and it’s difficult to find the clarity in the results descriptions.

Response: The data statistics used in this paper include each athlete's PFT score, percentage of PFT ranking, competition placements and points, and these are now descriped in the Section 2 in the revised manuscript. 

Results

- This part of the manuscript is divided into sections - the same structure could be adapted to describe statistical analysis in the Methods section.

Response: Thanks for your advice, however, we think this part could be kept because the description of this part of the statistics is not sufficient to support a subsection.

- line 107 Total points calculation should be moved to the methods section

Response: Thanks for your valuable comment, we have moved this part to the section 2.

- line 133-136 The title and description of the figure should be separated.

Response: Thanks for your suggestion. We have revised them accordingly.

- line 149 “quarterfinals” how this term relates to the term race? The descriptions in the methods section should clearly explain that.

Response: Athletes entering the quarter-finals means entering the top eight, and in 2020 NTCS means they can get points. In addition, from the quarterfinals to the semifinals, the top 4 of these 8 athletes are directly selected through the PFT rankings. Therefore, in order to avoid the bias caused by this competition system to study the influence of PFT results on athletes' competitive performance, we discussed entering the quarterfinals (section 3.1.2) and winning the first place in the quarterfinals (section 3.1.3) separately.

- line 183 “eight heavyweights'' should be changed into “weight categories” or “weight divisions”. Although standardEnglish is used, some terms in the text are misleading.

Response: Thanks for your advice, all the “heavyweight” are replaced by “weight category” in the revised manuscript.

- line 253 “quite good” should be changed to different, more objective term

Response: This statement has been revised.

- line 255 and Fig 5 names of participants were presented - did the researchers receive the consent to use the personal details? Prior to sharing human research participant data, authors should consult with an ethics committee to ensure data are shared in accordance with participant consent and all applicable local laws. Data sharing should never compromise participant privacy. It is therefore not appropriate to publicly share personally identifiable data on human research participants.

Response: Thanks for your kindly reminder, we have removed all the real names of the five elite athletes that appeared in the original manuscript.

- Knowing the rationale for the inclusion of each analysis and having a clear description of which participants, from which stage of competition took part in it, would have increased the understanding of the whole research design. I believe that each analysis was conducted with the purpose but neither methodological section nor the discussion explains their value and intention in an intelligible fashion.

Response: Thanks for your valuable comments. The main aim of this paper is to study the relationship between PFT results and taekwondo athletes' competitive performance by analyzing the results of four competitions in the 2020 NTCS. In order to study this issue, we first discussed the relationship between the athletes’ total scores in four competitions and their PFT results, and found that there is an exponential positive correlation between the total scores and PFT rankings (Section 3.1.1); we conducted research on the relationship between entering the quarterfinals (section 3.1.2) and winning first place in the semi-finals (section 3.1.3) and their PFTs, respectively, and found that athletes with relatively good PFT rankings have a better chance of achieving good results in the competition; then we carried out the individual PFT results of the athletes, analysis shows that 30-meter running is a key physical fitness index that determines the athlete's competition performance (section 3.2); finally we discussed the athletes' PFTs further in section 3.3.

Conclusions

- Part of this section should be added to the discussion part of the manuscript.

- Had the research questions been clearly and specifically formulated in the introduction or methods, the conclusions could have been more precise.

Response: Thanks for your valuable comments, we have enriched the discussion in the revised manuscript. As the last paragraph of the introduction declares, the question of our study is to figure out the relationship between PFT results and taekwondo athletes' competitive performance.

 

Reviewer #3:

- In summary, you need to used proper terminology with regards to Taekwondo. For example we do not use “race” in TKD but “match” or competition. there are other vocabulary that I highlighted for you.

Response: Thank you for pointing this out. We have addressed all the “race” in the manuscript into “competition”.

- You need to include the studies on the validity and reliability of the Physical Fitness Tests (PFT) used in this study. Who devised The scoring system used in your study and was it tested and reliable?

- Line 87, you need to explain each component of PFT separately with reference and validity and reliability scores.

- Line 96, what is the reference for this scoring?

Response: The PFT items, scoring criteria and test criteria are all decided by the NTCS event organising committee, including the Chinese Taekwondo Association and the Jiangsu Provincial Sports Bureau. (We have added this sentence to the revised manuscript, in Line 122-124). For the scoring criteria used in the 2020 NTCS PFT, we found no published literature to prove its reliability. On the other hand, as we used the rankings of the athletes' fitness tests in our analysis, we avoided, as far as possible, the bias caused by possible uncertainties in the scoring system. And the item set-up in 2020 NTFS PFT is similar to the German motor test, the US Army Air Forces PFT, the physical fitness scoring system for naval service personnel and the national student fitness test in China (Line 101-103 in the revised manuscript).

As Fig. 4 shows, the scale does not seem to clearly differentiate between the athletes' weighted squat, back endurance and abdominal endurance abilities (almost all athletes achieved a perfect score). This could be a potential source of bias in the results as there are two explanations, one being that the professional athletes did have good leg strength, abdominal endurance and back endurance, and the other being that the standard for a perfect score was not high enough. We have added this part of the discussion to the revised manuscript (Line 316-319).

- Your tables are very confusing and need to be redone. You have used too many tables and figures. you need to decrease the numbers and do not use table and figure to state the same data.

Response: Thanks for your suggestion, but we believe that every figure and every table in the manuscript is essential to our study. Firstly, Tables 1 and 2 give the scoring rule for the 2020 NTCS and scoring criteria for the PFT, respectively, which are the basis for the results that follow; Figure 1 shows the relationship between total points and PFT results; Tables 3 and 4 list the parameters and goodness of fit of the 18 fitted curves in Figure 1, which we believe are necessary for the analysis of the results; Figure 2 shows the PFT rankings of the top eight male athletes and top eight female athletes in the four competitions; Figure 3 summarises the probability of athletes with different PFT ranking percentages reaching the Top 8 in all four competitions, which is a further insight into Figure 2; Table 5 lists the number of times athletes with different rankings won the first place after reaching the Top 4, which is also beneficial for studying the relationship between competition performance and fitness of taekwondo athletes; Figure 4 shows the difference between the individual physical test scores of athletes who reached the Final 8 and those who did not; Figure 5 compares the PFT scores of the elite national team athletes with the average PFT scores of the athletes in their weight categories, again showing the importance of physical fitness to competitive performance.

 We think no two figures or tables state the same data. 

- Your discussion portion of your paper lacks the comparison with other studies and stating the reasons behind your finding in light of the literature. This is very crucial and essential for the success of a paper.

Response: Yes we totally agree with you that the comparison with other studies is very crucial for a research paper, actually we have worked hard to do this, for example, in Line 233, Line 257 and Line 305. However, as there are not many relevant studies, more comparisons seem very challenging. 

- One last but most concerting is the ethical issues of this study. It is unethical to include the name of athletes or any identifying data about the athlete in a scientific peer reviewed paper

Response: Thanks for your kindly reminder, we have removed the names of any athletes involved in the paper.

- Line 99, was this base on PFT scores or TKD competition?

- Line 101, why? This could be a source of bias.

Response: This is based on PFT scores. Because according to the rule of 2020 NTCS, after the top 8 players are determined, the top 4 of these 8 athletes in term of PFT ranking will go directly to semi-finals, but not based on TKD competition. We acknowledge that this system may create some bias, mainly in that it is possible that athletes who do not do well in the PFT also have the opportunity to advance in the 8-in-4. This bias may be responsible for the exponential, rather than linear, relationship illustrated in Figure 1 when discussing the relationship between the final total score and PFT result (discussed in Line 188-192). Therefore, in order to avoid this bias, the relationship between " Entering the quarterfinals" (section 3.1.2) and " Winning in semifinals" (section 3.1.3) and the PFT results are analysed and discussed separately. 

- Line 110, is this your idea or has it been published and validated?

Response: The organising committee of the 2020 NTCS gave the rules for points, as shown in Table 1, with 100 points for first place, 60 points for second place, 36 points for third place and 21.6 points for fourth place. When we trace the exponential relationship between total points and the PFT results (section 3.1.1), we found that the scoring rule of 2020 NTCS can also be expressed exactly as an exponential function. We have therefore summarised the scoring rule of 2020 NTCS as a exponential function, which can be easier for other tournaments to refer to and generalise, and it is easy to verify this formula.

- Line 187, this s very confusing. Was there only one competition? Are you referring to each match and each race? You need to clarify by winning the competition do you mean the athlete got gold medal?

- Line 189, this is very confusing which one is the athlete's rank in PFT and which one is their rank in the competition?

Response: Perhaps the introduction of the 2020 NTCS was not presented clearly enough before, which led to the confusion here. We have now revised the writting of the 2020 NTCS competition rules and PFT rules in the methods section and hope to make the results and discussion in this section clearer.

 As the 8-in-4 is directly determined by the PFT results and not the competition results (as discussed earlier, there may be some bias here), in order to examine the relationship between athletes' competition performance and fitness levels, we need to devide the discussion into two parts, one is that presented in section 3.1.2, the other one is “whether athletes with better PFT results have a higher probability of winning the first place after reaching the semifinal” as discussed in this section.

 There are 4 competitions in 2020 NTCS (first column in Table 5), in the case of the male athelets, for example, each competition produced eight players who were first place, as there were eight weight categories. For these 8 first place athletes, it is helpful to count their PFT ranking among the top four athletes in their respective weight categories. Assuming there is no relationship between competition performance and fitness, the number of times the No. 1, No. 2, No. 3 and No. 4 ranked athletes in the PFT have won first place in a competition should all be 2, however, actually it is 4, 1, 2, and 1 (as the first line of number in Table 5), which means that the athlete who ranks first in the physical test among these four is more likely to win first place in the competition.

- Line 218, were these differences statistically significant? Please state.

Response: This phrase was considered too colloquial and also redundant, so we deleted it.

- Line 236, how are these compared to other sports and other references. In your discussion you need to compare your results to other papers and state the reasons.

Response: The comparisons we make here are between the athletes studied in this paper, as the exact same fitness test criteria are not used in the other literature, we are unable to make comparisons with other sports and other references. In fact this conclusion can already be drawn from Figure 5.

- Line 254, higher than who and what is you reference?

Response: Thanks for your comment, we have reorganized this sentence, now it’s in the Line 286-290 in the revised manuscipt.

---

## [Decision Letter · Decision Letter 1]

11 Feb 2022

PONE-D-21-18355R1The relationship between physical fitness and competitive performance of Taekwondo athletes

PLOS ONE

Dear Dr. He,

Thank you for submitting your manuscript to PLOS ONE. After careful consideration, we feel that it has merit but does not fully meet PLOS ONE’s publication criteria as it currently stands. Therefore, we invite you to submit a revised version of the manuscript that addresses the points raised during the review process.

We look forward to receiving your revised manuscript.

Kind regards,

Krzysztof Durkalec-Michalski, Ph.D

Academic Editor

PLOS ONE

Additional Editor Comments:

The comments of the reviewer should be fully complied with.Descriptions of figures are unclear and it is difficult for the reader to fully accurately interpret the presented data - the quality of figure descriptions and their descriptions under the figures should be clearly improved.Lines 173-174 - Remove this torch "high-flown" stylistically sentence (Remove this torch "high-flown" stylistically sentence (“Potential possible causes for this phenomenon will be discussed later”).  At the same time, make it clear in the discussion that you are referring to these observations from the results.Consider adding legends under the tables to help the reader understand the tables.Tables should be properly edited to make it easier to follow (breaking descriptions into lines makes it difficult.The name of table 4 (description) is unreliable. The table should be fully and properly named so as not to raise doubts and not to send the reader to look for elsewhere.Statements such as "This feature is more obvious for" should be limited - statements "obvious" are not entirely accurate in the discussed contexts.Again - the description in tables, figures and in the text must allow for an authoritative and unambiguous assessment of what they refer to and what they indicate. At the moment, only the reader who has read the work very carefully, are they clear. It is necessary to precisely and clearly describe what and what these figures indicate, with whom and when.Lines 256-257 – “…athletes are pretty good.” – reword this using more scientific language and past time.Line 258 – avoid using abbreviations  “didn’t”.Consider whether in every context in the work naming athletes who train taekwondo as "players" is reliable - I have some concerns here.The discussion is quite poor when it comes to comparisons with other studies.A large part of the conclusions and limitations is basically a discussion - please move it to the previous section. Limit your conclusions to what has actually been shown.

Reviewers' comments:

Reviewer's Responses to Questions

**Comments to the Author**

1. If the authors have adequately addressed your comments raised in a previous round of review and you feel that this manuscript is now acceptable for publication, you may indicate that here to bypass the “Comments to the Author” section, enter your conflict of interest statement in the “Confidential to Editor” section, and submit your "Accept" recommendation.

Reviewer #1: All comments have been addressed

Reviewer #3: (No Response)

2. Is the manuscript technically sound, and do the data support the conclusions?

Reviewer #1: Yes

Reviewer #3: No

3. Has the statistical analysis been performed appropriately and rigorously? 

Reviewer #1: Yes

Reviewer #3: No

4. Have the authors made all data underlying the findings in their manuscript fully available?

Reviewer #1: Yes

Reviewer #3: Yes

5. Is the manuscript presented in an intelligible fashion and written in standard English?

Reviewer #1: Yes

Reviewer #3: No

6. Review Comments to the Author

Reviewer #1: Starting off with a appraise, the Authors managed to address most of the issues pinpointed by the reviewers, in an orderly and clear manner. Necessary addendums as well as removals have substantially enriched the text as a whole. As stated previously, some of the less fortunate terms were present, but the Authors managed to correct them this time.

Without further ado, I shall voice out the most important aspects of the current state of the article:

1) The Description of the PFT test…

From where I am standing, the addendums in lines 101-103 clear out any doubts.

2) What kind of equipment was used…

Necessary corrections have been applied. It is perfectly understandable, that some of the more “mundane” and “less spectacular” items may have been manufactured so long ago, that the detailed information may have been next to impossible to obtain.

3) Addressing the phrases from lines 85-86…

The abovementioned issue has been addressed accordingly.

4) After careful analysis of the attached citations…

The Authors did underline the separation from biomonitoring towards a more macro-athletic approach. The limitations were provided in the renewed version of the manuscript.

5) Why did the research team opt not to carry out an analysis…

The most hampering limitations were added by the research team in lines 313-316. The provided reasons are understandable and I strongly encourage taking them into consideration, which may finally result in obtaining data viable for an analysis of covariance. Maybe it was not aired specifically enough – no World/Asian/Olympic/Cup medalists whatsoever in said group? Just a general quota would suffice – i.e. 11 gold/22 silver/33 bronze etc.

6) As far as I am concerned, the Authors should specifically underline…

The appropriate lines and addendums were administered by the Authors.

7) Multiple times did the Authors underline the benefits of introducing the PFT testing…

In the light of multiple alterations carried out in the text, my initial doubts have been dispelled. The percentages presented in Figure nr.2 are clear. Still, I strongly suggest tackling the issue in future research regarding thresholds in the PFT test.

Reviewer #3: Thank you for your revisions. It reads much better.

However, you have not provided any evidence/references on the individual tests of the PFT reliability, validity or any reference on the way they were performed in your study or the normal values of each test in comparison to TKD athletes in your study. You have not explain the utility of each test and significance of each test physiologically and why they should be included in the battery of the tests. You also have not provided any p values for the differences you mentioned in various athlete groups, male versus female to show statistical significance.

As such, unfortunately in my opinion your manuscript in not publishable in its present form.

7. PLOS authors have the option to publish the peer review history of their article (what does this mean?). If published, this will include your full peer review and any attached files.

Reviewer #1: No

Reviewer #3: No

---

## [Author Response · Author response to Decision Letter 1]

20 Mar 2022

Responses to the Editor and Reviewers

Dear Editor Durkalec-Michalski, 

Thanks for your valuable comments. In the following we answer each specific point (in blue). 

Descriptions of figures are unclear and it is difficult for the reader to fully accurately interpret the presented data - the quality of figure descriptions and their descriptions under the figures should be clearly improved.

Thanks for your insight into this. We have improved the description of the figures, both in the text and in the legend, especially the descriptions of Figs 1 to 4 have been significantly modified. The missing legend has been added to Fig 2 (red ladders). We hope these modifications could make the figures easier to understand.

Lines 173-174 - Remove this torch "high-flown" stylistically sentence (“Potential possible causes for this phenomenon will be discussed later”). At the same time, make it clear in the discussion that you are referring to these observations from the results.

Sorry for the colloquial expressions, this sentence has been removed. And we have made it clear that the discussion was based on the results in the previous section, for example, in Line 303 of the revised manuscript we stated that "This may explain the results in section 3.1.1."

Consider adding legends under the tables to help the reader understand the tables.

Thanks for the valuable suggestion, we have now added some legends to make the table easier to understand, e.g., Table 3 and Table 5. 

Tables should be properly edited to make it easier to follow (breaking descriptions into lines makes it difficult.

We have rearranged Tables 3 and 4, which may not be easy to follow as mentioned by the editor. 

The name of table 4 (description) is unreliable. The table should be fully and properly named so as not to raise doubts and not to send the reader to look for elsewhere.

This too brief description has now been reformulated to "Table 4. The fitting coefficient (a and b), the Goodness of Fit (R2) and the number of athletes (N) in Fig. 1d-e. "

Statements such as "This feature is more obvious for" should be limited - statements "obvious" are not entirely accurate in the discussed contexts.

Thank you for pointing this out, we have replaced the word "obvious" to "pronounced" to make it more academic.

Again - the description in tables, figures and in the text must allow for an authoritative and unambiguous assessment of what they refer to and what they indicate. At the moment, only the reader who has read the work very carefully, are they clear. It is necessary to precisely and clearly describe what and what these figures indicate, with whom and when.

We agree with the editor that the description of the figures and tables should be clear and easy to understand, and we have modified almost every figure and table description in the text accordingly for this purpose. 

Lines 256-257 – “…athletes are pretty good.” – reword this using more scientific language and past time. (Line 239?)

Thanks for the advice, we have now reformulated this sentence to read "This means that … of these professional Taekwondo athletes all achieved the standard of excellence set by the General Administration of Sport of China."

Line 258 – avoid using abbreviations “didn’t”. (Line 240?)

This abbreviation has been revised.

Consider whether in every context in the work naming athletes who train taekwondo as "players" is reliable - I have some concerns here.

We agree with the editor that "players" may be not a appropriate word here, we have replaced this word with "athletes" in the revised manuscript.

The discussion is quite poor when it comes to comparisons with other studies.

In this revised version, we have made significant modifications in the section 3 (Results and discussion) according to the editor and the reviewer’s suggestions, which, we hope, has improved the disussion and made it more clear. Some major modifications, including but not limited to Lines 159-162, Lines 208-211, Lines 272-280 and Lines 286-288.

A large part of the conclusions and limitations is basically a discussion - please move it to the previous section. Limit your conclusions to what has actually been shown.

Thanks for pointing out this, we agree with the editor that that some sentences should be placed in the discussion section, specifically, we have moved Lines 313-316 in the previous manuscript to Lines 208-211 in the revised version, moved Lines 316-319 in the previous manuscript to Lines 286-288 in the revised version. Some other modifications in the conclusions could also be clear seen in the tracked version of manuscript.

Reviewer #1: Starting off with a appraise, the Authors managed to address most of the issues pinpointed by the reviewers, in an orderly and clear manner. Necessary addendums as well as removals have substantially enriched the text as a whole. As stated previously, some of the less fortunate terms were present, but the Authors managed to correct them this time.

Thanks for your overall comments. In the following we answer each specific point (in blue).

Without further ado, I shall voice out the most important aspects of the current state of the article:

1) The Description of the PFT test…

From where I am standing, the addendums in lines 101-103 clear out any doubts.

Thank you, further modifications have also been made in this revision, for example, in Lines 105-109, Lines 110-111 and Lines 125-126 in the revised manuscript. 

2) What kind of equipment was used…

Necessary corrections have been applied. It is perfectly understandable, that some of the more “mundane” and “less spectacular” items may have been manufactured so long ago, that the detailed information may have been next to impossible to obtain.

Thank you for your understanding, we agree with the reviewer that these informations are important for other researchers. 

3) Addressing the phrases from lines 85-86…

The abovementioned issue has been addressed accordingly.

4) After careful analysis of the attached citations…

The Authors did underline the separation from biomonitoring towards a more macro-athletic approach. The limitations were provided in the renewed version of the manuscript.

Thanks for your comment, although this more macro-athletic approach has its limitations, we think it is still meaningful as it is at least a complement to the bioassay approach.

5) Why did the research team opt not to carry out an analysis…

The most hampering limitations were added by the research team in lines 313-316. The provided reasons are understandable and I strongly encourage taking them into consideration, which may finally result in obtaining data viable for an analysis of covariance. Maybe it was not aired specifically enough – no World/Asian/Olympic/Cup medalists whatsoever in said group? Just a general quota would suffice – i.e. 11 gold/22 silver/33 bronze etc.

Thanks for the reviewer’s suggestions, and we understand the concerns of the reviewer. We are sorry that we are not in a position to adopt this recommendation as we failed to obtain these additional information. However, we believe that these additional information, although beneficial for doing an analysis of covariance, will not affect the main findings of this study. Athletes' competitive performance is certainly also related to other factors such as taekwondo skills, mental fitness and competition experience (which would be related to whether or not they have previously won honours), but as our study only discusses the relationship between athletes' physical test scores and their competitive performance, the current data and statistics should be sufficient to justify the conclusions of this paper. 

6) As far as I am concerned, the Authors should specifically underline…

The appropriate lines and addendums were administered by the Authors.

7) Multiple times did the Authors underline the benefits of introducing the PFT testing…

In the light of multiple alterations carried out in the text, my initial doubts have been dispelled. The percentages presented in Figure nr.2 are clear. Still, I strongly suggest tackling the issue in future research regarding thresholds in the PFT test.

Thanks for you valuable suggestion, some of the limitations in this study will be discussed and improved upon in more detail in subsequent studies. 

Reviewer #3: Thank you for your revisions. It reads much better.

Thank you for your contribution to the improvement of this manuscript. In the following we answer each specific point (in blue).

However, you have not provided any evidence/references on the individual tests of the PFT reliability, validity or any reference on the way they were performed in your study or the normal values of each test in comparison to TKD athletes in your study. You have not explain the utility of each test and significance of each test physiologically and why they should be included in the battery of the tests. 

Thanks for your valuable comments and sorry for the limited description of the physical fitness test in previous manuscripts. We have added the meanings of this battery of physical tests to the text, which reads "This battery of tests can be considered as an indicator stand for various aspects of an athlete's athletic ability, including agility, explosive power, strength, aerobic capacity and anaerobic capacity, which has been viewed as a multidimensional structure that reflects motor performance ability [27]", and indicated the utility and physiological significance of each individual test, for example, "Weight-bearing squat is used to test the strength of the quadriceps, gluteus maximus and other lower limb muscles [28]. ", "The 30-meter sprint test is designed to measure speed ability [23]", "The 3000 m run is a test of aerobic endurance [23]", whereas abdominal and back endurance are measured in a similar way to [29] (but not exactly the same), the specific details of each individual test are set out in the revised manuscript (Lines 110-133).

Unfortunately no reference was found to demonstrate what normal values should be achieved by TKD athletes in these individual tests. However, we believe that since the subject of our study in this paper is to compare the relationship between competitional performance and physical test scores among these professional TKD athletes who participated the 2020 NTCS, and not to compare them with other athletes, the absence of the "normal values" seems does not affect the conclusions. 

You also have not provided any p values for the differences you mentioned in various athlete groups, male versus female to show statistical significance.

Sorry the unclear writing, we have tried to clarify this in the previous manuscript in Lines 142-143, which reads "All p-values in this article are less than 0.001, and will not be marked hereafter", however, it seems not clear enough especially as the reader tends to forget this when reading later. We have now removed the Lines 142-143 in the previous version of manuscript and instead, we have clarified p-values where they occur. For example, in Line 204 and Line 299.

---

## [Decision Letter · Decision Letter 2]

11 Apr 2022

PONE-D-21-18355R2The relationship between physical fitness and competitive performance of Taekwondo athletesPLOS ONE

Dear Dr. He,

Thank you for submitting your manuscript to PLOS ONE. After careful consideration, we feel that it has merit but does not fully meet PLOS ONE’s publication criteria as it currently stands. Therefore, we invite you to submit a revised version of the manuscript that addresses the points raised during the review process.

We look forward to receiving your revised manuscript.

Kind regards,

Krzysztof Durkalec-Michalski, Ph.D

Academic Editor

PLOS ONE

Journal Requirements:

Reviewers' comments:

Reviewer's Responses to Questions

**Comments to the Author**

1. If the authors have adequately addressed your comments raised in a previous round of review and you feel that this manuscript is now acceptable for publication, you may indicate that here to bypass the “Comments to the Author” section, enter your conflict of interest statement in the “Confidential to Editor” section, and submit your "Accept" recommendation.

Reviewer #3: All comments have been addressed

2. Is the manuscript technically sound, and do the data support the conclusions?

Reviewer #3: Yes

3. Has the statistical analysis been performed appropriately and rigorously? 

Reviewer #3: Yes

4. Have the authors made all data underlying the findings in their manuscript fully available?

Reviewer #3: Yes

5. Is the manuscript presented in an intelligible fashion and written in standard English?

Reviewer #3: No

6. Review Comments to the Author

Reviewer #3: Thank you for the revisions. Allmy concerns have been addressed. However, there some grammatical and other minor errors that require revisions as follow:

line 25: specifically in increase of explosive power. pllease change to "specially enhancing the explosive power."

Line 57, "the PFTs performed are all conducted". change to: "the PFTs performed were all conducted"

Line 61, "Aware of the important role"change to "Being aware of..."

Line 76, "The 2020 NTCS contains four competitions, two PFTs and used a scoring system.", change to, "The 2020 NTCS contained four competitions, two PFTs and a scoring system."

Line 116, "endurance of the abdominal muscles, the test subject lies prone on a jump box with the"

Should be, "endurance of the abdominal muscles, the test subject lies supine on a jump box with the"

Line 120-121, "back muscle endurance is the same as abdominal muscle endurance, except that the athlete lies on the jumping box" should be:"back muscle endurance is the same as abdominal muscle endurance, except that the athlete lies prone on the jumping box"

7. PLOS authors have the option to publish the peer review history of their article (what does this mean?). If published, this will include your full peer review and any attached files.

Reviewer #3: No

---

## [Author Response · Author response to Decision Letter 2]

11 Apr 2022

Responses to the reviewers

Reviewers' comments to the Author:

Reviewer #3: Thank you for the revisions. All my concerns have been addressed. However, there some grammatical and other minor errors that require revisions as follow:

Thanks for your careful checking. We have corrected all the following typos and grammatical errors the reviewer raised.

line 25: specifically in increase of explosive power. please change to "specially enhancing the explosive power."

Modified.

Line 57, "the PFTs performed are all conducted". change to: "the PFTs performed were all conducted"

Corrected.

Line 61, "Aware of the important role"change to "Being aware of..."

Corrected.

Line 76, "The 2020 NTCS contains four competitions, two PFTs and used a scoring system.", change to, "The 2020 NTCS contained four competitions, two PFTs and a scoring system."

Modified.

Line 116, "endurance of the abdominal muscles, the test subject lies prone on a jump box with the"

Should be, "endurance of the abdominal muscles, the test subject lies supine on a jump box with the"

Corrected.

Line 120-121, "back muscle endurance is the same as abdominal muscle endurance, except that the athlete lies on the jumping box" should be:"back muscle endurance is the same as abdominal muscle endurance, except that the athlete lies prone on the jumping box"

Corrected.

---

## [Decision Letter · Decision Letter 3]

14 Apr 2022

The relationship between physical fitness and competitive performance of Taekwondo athletes

PONE-D-21-18355R3

Dear Dr. Lumin He,

We’re pleased to inform you that your manuscript has been judged scientifically suitable for publication and will be formally accepted for publication once it meets all outstanding technical requirements.

Kind regards,

Krzysztof Durkalec-Michalski, Ph.D

Academic Editor

PLOS ONE

Reviewers' comments:

Reviewer's Responses to Questions

**Comments to the Author**

1. If the authors have adequately addressed your comments raised in a previous round of review and you feel that this manuscript is now acceptable for publication, you may indicate that here to bypass the “Comments to the Author” section, enter your conflict of interest statement in the “Confidential to Editor” section, and submit your "Accept" recommendation.

Reviewer #3: All comments have been addressed

2. Is the manuscript technically sound, and do the data support the conclusions?

Reviewer #3: Yes

3. Has the statistical analysis been performed appropriately and rigorously? 

Reviewer #3: Yes

4. Have the authors made all data underlying the findings in their manuscript fully available?

Reviewer #3: Yes

5. Is the manuscript presented in an intelligible fashion and written in standard English?

Reviewer #3: Yes

6. Review Comments to the Author

Reviewer #3: I would like to thank the Authors for addressing all my concerns. The manuscript at the present form is readable and publishable.

Kind Regards

7. PLOS authors have the option to publish the peer review history of their article (what does this mean?). If published, this will include your full peer review and any attached files.

Reviewer #3: No

---

## [Editor Report · Acceptance letter]

8 Jun 2022

PONE-D-21-18355R3 

The relationship between physical fitness and competitive performance of Taekwondo athletes 

Dear Dr. He:

I'm pleased to inform you that your manuscript has been deemed suitable for publication in PLOS ONE. Congratulations! Your manuscript is now with our production department. 

Kind regards, 

on behalf of

Dr. Krzysztof Durkalec-Michalski 

Academic Editor

PLOS ONE